# Melatonin Improves Skeletal Muscle Structure and Oxidative Phenotype by Regulating Mitochondrial Dynamics and Autophagy in Zücker Diabetic Fatty Rat

**DOI:** 10.3390/antiox12081499

**Published:** 2023-07-27

**Authors:** Diego Salagre, Enrique Raya Álvarez, Cruz Miguel Cendan, Samira Aouichat, Ahmad Agil

**Affiliations:** 1Department of Pharmacology, BioHealth Institute Granada (IBs Granada), Neuroscience Institute (CIBM), School of Medicine, University of Granada, 18016 Granada, Spain; dsalagre@ugr.es (D.S.);; 2Department of Rheumatology, University Hospital Clinic San Cecilio, 18016 Granada, Spain

**Keywords:** melatonin, mitochondrial fission-fusion, autophagy, oxidative stress, skeletal muscle, red vastus lateralis, muscle fiber composition, diabesity, ZDF rat

## Abstract

Obesity-induced skeletal muscle (SKM) inflexibility is closely linked to mitochondrial dysfunction. The present study aimed to evaluate the effects of melatonin on the red vastus lateralis (RVL) muscle in obese rat models at the molecular and morphological levels. Five-week-old male Zücker diabetic fatty (ZDF) rats and their age-matched lean littermates (ZL) were orally treated either with melatonin (10 mg/kg body weight (BW)/24 h) (M–ZDF and M–ZL) or non-treated (control) (C–ZDF and C–ZL) for 12 weeks. Western blot analysis showed that mitochondrial fission, fusion, and autophagy were altered in the C-ZDF group, accompanied by reduced SIRT1 levels. Furthermore, C-ZDF rats exhibited depleted ATP production and nitro-oxidative stress, as indicated by increased nitrites levels and reduced SOD activity. Western blotting of MyH isoforms demonstrated a significant decrease in both slow and fast oxidative fiber-specific markers expression in the C-ZDF group, concomitant with an increase in the fast glycolytic fiber markers. At the tissue level, marked fiber atrophy, less oxidative fibers, and excessive lipid deposition were noted in the C-ZDF group. Interestingly, melatonin treatment partially restored mitochondrial fission/fusion imbalance in the RVL muscle by enhancing the expression of fission (Fis1 and DRP1) markers and decreasing that of fusion (OPA1 and Mfn2) markers. It was also found to restore autophagy, as indicated by increased p62 protein level and LC3BII/I ratio. In addition, melatonin treatment increased SIRT1 protein level, mitochondrial ATP production, and SOD activity and decreased nitrites production. These effects were associated with enhanced oxidative phenotype, as evidenced by amplified oxidative fiber-specific markers expression, histochemical reaction for NADH enzyme, and muscular lipid content. In this study, we showed that melatonin might have potential therapeutic implications for obesity-induced SKM metabolic inflexibility among patients with obesity and T2DM.

## 1. Introduction

Obesity is a strong risk factor for Type 2 Diabetes Mellitus (T2DM) and numerous other comorbidities, and its prevalence is expected to increase dramatically over the coming decades [1]. Insulin resistance is the hallmark of obesity and T2DM and is the core of pathophysiological characteristics of insulin-sensitive tissues, including skeletal muscle (SKM) [2,3,4]. The pathophysiological concept underlying the diabesity (obesity and T2D) condition is termed lipotoxicity, that is, ectopic accumulation of lipids and their species in metabolic tissues, particularly SKM, negatively affecting their metabolic function [2].

In mammals, SKM is composed of a heterogeneous collection of muscle fiber types, as they comprise slow-twitch oxidative type I fibers, intermediate-twitch oxidative type IIa fibers, fast-twitch oxidative glycolytic type IIx fibers, and fast-twitch glycolytic type IIb. Type I fibers contain numerous mitochondria and principally rely on oxidative phosphorylation for ATP production. On the other hand, type II fibers have fewer mitochondria and generate energy mainly through glycolytic metabolism. Previous studies in humans and rodents established that SKM is endowed with significant phenotypic plasticity and adaptability under various conditions, including exercise and obesity. For instance, obesity instigates a shift in the composition of muscle fibers, favoring a transition towards a glycolytic phenotype, thereby leading to ectopic lipid accumulation and insulin resistance in SKM, ultimately hastening the progression of metabolic abnormalities [5,6].

SKM represents more than 50% of the total body weight (BW) in adults and is responsible for 70 to 80% of insulin-stimulated glucose uptake in the body of healthy adults, and its metabolic plasticity allows a rapid response to the cellular energy demand of the whole body thanks to its large number of mitochondria [7,8,9,10]. SKM is therefore crucially important for its role in maintaining whole-body glucose disposal, and alterations in its metabolic function play an important role in the pathogenesis of T2DM [5,9]. In obesity and T2D, the SKM displays a phenotype characterized by lower fiber oxidative capacity and muscular lipid deposition associated with lower mitochondrial content, which is related to insulin resistance and muscle mass loss [5,11].

A key component of maintaining SKM metabolic plasticity and function is the mitochondria, which are vital organelles in fibers, responsible for controlling SKM homeostasis through regulating metabolic process, ATP production, proteostasis, oxidative statue, and Ca^2+^ handling [12]. Mitochondrial dysfunction has been described as a pivotal metabolic defect in obesity, contributing to the accumulation of lipotoxic lipids species and insulin resistance [13]. Structurally, the maintenance of mitochondrial homeostasis is tightly governed by mitochondrial network dynamics, encompassing mitochondrial fission, mitochondrial fusion, and mitophagy [14]. Abundant evidence has demonstrated that mitochondrial network dynamic dysfunction in SKM is a key mechanism mediating insulin resistance and accumulation of lipotoxic lipid species that characterize the diabesity state [15,16,17]. This has been reported in patients with obesity, insulin resistance, and T2DM, as well as in several animal models [18,19,20,21].

Mitochondrial fission and fusion are tightly regulated by several protein complexes. Mitofusins 2 (Mfn2) and optic atrophy protein type 1 (OPA1) mediate mitochondrial fusion, whereas dynamin-related protein (DRP1) and fission 1 protein (Fis1) are considered to be key proteins in mitochondrial fission [22].

Autophagy is mainly regulated by p62, also known as sequestosome 1 (SQSTM1), which binds ubiquitinated proteins leading to the formation of autophagosomes for subsequent degradation [23,24]. One of the traditional pathways of autophagy induction is through the SIRT1, a member of the Sirtuins family of NAD+-dependent protein deacetylases seemingly involved in regulating numerous mitochondrial metabolic and dynamic pathways, ageing, inflammation, and apoptosis [25,26,27,28]. Studies in rodents have demonstrated that impaired SIRT1 expression reduced autophagic flux, leading to lipid accumulation, lipotoxicity, and glucose intolerance in the SKM and liver [29,30].

Recently, there has been evidence in animal and human studies that the mitochondrial dynamic process is the basis for mitochondrial functions, ATP production, oxidative stress balance, metabolic flexibility, and insulin sensitivity [11,31,32,33,34]. Therefore, maintaining the mitochondrial dynamic balance in SKM is crucially essential to meet the energy demands of the body. Obesity has been shown to deteriorate the mitochondrial network, resulting in mitochondrial dysfunction and a pro-oxidant state [35,36,37,38]. Studies in rodents and humans have shown that obesity impaired the mitochondrial fusion–fission balance leading to reduced oxidative phosphorylation, ATPase activity, and antioxidant defenses [18,39,40,41,42,43,44]. The above findings highlighted the relevance of targeting mitochondrial dynamics towards optimal mitochondrial function, glucose homeostasis, insulin sensitivity, and thereby diabesity control.

Melatonin is an indolamine produced mainly by the pineal gland during the dark phase but also locally in other tissues [45,46]. In addition to regulating circadian rhythms, it has antioxidant, anti-inflammatory, and energy balance regulating effects, which limit obesity, insulin resistance, hyperglycemia, and dyslipidemia [47,48,49,50,51,52] in the same rat strain as shown in previous studies of our research group [53]. Given its extraordinary properties and the fact that it is a well-recognized mitochondria-targeting agent, melatonin could be an effective approach for regulating SKM mitochondrial dynamics under obese and diabetes status [54,55,56,57]. Treatment with melatonin has been shown to activate the SIRT1 signaling and restore autophagy and mitochondrial dynamics in obesity-induced hepatic steatosis [58,59]. Additionally, in vitro studies have shown that melatonin increased autophagy in a dose-dependently manner in C2C12 myoblast by activating LC3bII and preventing mitochondrial dysfunction and cell death [60,61]. However, more studies are needed for a better understanding of the molecular pathway involved in the SKM mitochondrial dynamic regulation.

The ZDF rat is a widely used animal model due to its similar pathogenesis and evolution with human T2DM [62,63]. Thus, this study aimed to determine whether chronic oral melatonin treatment could restore the diabesity-induced mitochondrial dynamic imbalance in ZDF rats and, if so, whether it would reflect on the mitochondrial oxidative/nitrosative status and induce SKM tissue remodeling.

## 2. Materials and Methods

### 2.1. Reagents

All utilized reagents were of the utmost purity attainable. Melatonin was acquired from reputable sources, including Sigma-Aldrich (Madrid, Spain).

### 2.2. Ethics Approval

The animal experimentation conducted in this study adhered to the protocols approved by the Ethical Committee at the University of Granada (Granada, Spain), under the reference number 23/06/2021/096-CEEA. Furthermore, the study was carried out in accordance with the animal care and protection guidelines of the European Union.

### 2.3. Animals and Experimental Protocol

Five-week-old male Zücker diabetic fatty rats (ZDF; *fa*/*fa*; *n* = 16) and their age-matched male lean littermates (ZL; *fa*/–; *n* = 16) were obtained from Charles River Laboratory (Charles River Laboratories, SA, Barcelona, Spain). Rats were housed two per plastic cage in an animal facility with controlled conditions of temperature (28–30 °C) and relative humidity (30–40%) and a 12 h light/dark cycle (lights on at 07:00 a.m.) and had ad libitum access to tap water and Purina 5008 rat chow (protein 23%, fat 6.5%, carbohydrates 58.5%, 142 fiber 4%, and ash 8%; Charles River Laboratories, SA, Barcelona, Spain). After a 4-day adaptation period, ZL and ZDF rats were randomly divided into four groups (*n* = 8 per group): the melatonin-treated groups (M-ZDF and M-ZL) received melatonin in their drinking water at a dose of 10 mg/kg BW/24 h for 12 consecutive weeks, and the control groups (C-ZDF and C-ZL) received vehicle only at the same dosage volume and treatment duration as the treated groups. Melatonin was dissolved in 70% ethanol and then diluted in the drinking water to a final concentration of 0.066% (*w*/*v*). Fresh melatonin and vehicle solutions were prepared daily, and the dosing concentration of melatonin was calculated daily over the entire experimental period, based on the individual rat’s BW on each day of dosing. BW and Fluid intake were recorded daily to determine accurately the treatment dose throughout the experiment and ensure an effective dose of 10 mg/kg BW of melatonin. Feed intake was monitored daily. All drinking water bottles were covered with aluminum foil to protect solutions against the light. At the conclusion of the experiment, the rats were euthanized using sodium thiobarbital (thiopental) anesthesia, and the deep portion (red) of vastus lateralis (RVL) was quickly isolated by dissection from each rat and then weighted. Half of the collected RVL sample was oriented to obtain transverse sections and then either frozen-fixed in OCT mounting media for microscopic analysis or immediately stored in liquid nitrogen for further processing as described subsequently.

### 2.4. Total Protein Extraction and Western Blotting

Proteins were isolated from about 500 mg of RVL samples in RIPA Lysis Buffer (150 mM NaCl, 50 mM Tris-HCl, 1% Nonidet P-40, 0.5% sodium deoxycholate, and 0.1% sodium dodecyl sulfate (SDS)) with 1% Triton X-100 and 1% protease inhibitor cocktail (Thermo Fisher Scientific, Waltham, MA, USA). The samples were homogenized using a Teflon pestle while ensuring that the temperature was maintained at 4 °C throughout all procedures. Homogenates were then centrifuged at 13,000× *g* for 15 min at 4 °C. The clear supernatant was carefully transferred to a new microcentrifuge, and aliquots of total protein lysate were stored at −80 °C. Quantification of total isolated proteins of samples was determined by the Bradford method. Bovine serum albumin (BSA) was used as a standard.

The protein extracts (50 µg protein) were subjected to SDS-polyacrylamide gel electrophoresis (SDS-PAGE) for separation and analysis, and running buffer containing tricine (0.1 M Tris-HCl, 0.1% SDS, 0.1 M Tricine) was used for a better separation of the bands. The gels were transferred to a nitrocellulose membrane (Bio-Rad Trans-Blot SD, Bio-Rad Laboratories, Hercules, CA, USA), and then the membranes were blocked for 1 h at room temperature using Blocking Buffer (Phosphate Buffer Saline (PBS); 137 mM NaCl, 2.7 mM KCl, 10 mM Na_2_HPO_4_, and 1.8 mM KH_2_PO_4_, containing 1% Tween-20 and 5% non-fat dry milk). Following this, overnight incubation at 4 °C of blots was performed using primary antibodies raised in rabbit against OPA1 (cat#SAB-5700860; Sigma-Aldrich), DRP1 (cat#SAB-5700783; Sigma-Aldrich), Mfn2 (cat#SAB210801; Sigma-Aldrich), Fis1 (cat#HPA017430; Sigma-Aldrich), p62/SQSTM1 (cat#SAB570084; Sigma-Aldrich), LC3b (cat#SAB5700049; Sigma-Aldrich), SIRT1 (cat#SAB570109; Sigma-Aldrich), and MyH7 (cat#SAB2106550; Sigma-Aldrich) or in mouse against MyH1 (cat#MBS252930; MyBioSource), MyH2 (cat#MABT840; Sigma-Aldrich), MyH4 (cat#14-6503-82; Thermo Scientific) at 1:1000 dilutions in Blocking Buffer. Primary anti-β-actin antibody generated in mouse (cat#SC-81178; Santa Cruz Biotechnology, Dallas, TX, USA) was used as a loading control to confirm equal protein loading at 1:1000 dilution. After overnight incubation, the membranes were washed three times for 15 min in PBS containing 1% Tween-20 (PBS-T) to remove unbound primary antibodies and then incubated for 2 h at room temperature with respective horseradish peroxidase (HRP)-conjugated secondary antibodies (Sigma-Aldrich, Madrid, Spain) at 1:2000 dilution. After washing the membranes three times for 15 min in the PBS-T, immunoreactive proteins were revealed by enhanced chemiluminescence using the Clarity Western ECL commercial kit (BioRad, Santiago de Compostela, Spain) following the manufacturer’s instructions. The signal intensities were recorded with Kodak Image Station 4000 MM Pro Molecular Imaging and quantitatively analyzed using ImageJ 1.33 software (National Institutes of Health, Bethesda, MD, USA). To ensure accuracy, the results were normalized to β-actin, and all experiments were conducted in triplicate to account for reproducibility and reliability.

### 2.5. Cytosolic and Mitochondrial Fractions Preparation

The cytosolic and mitochondrial fractions were extracted from RVL samples by a modified differential centrifugation protocol that has the advantage of preserving the integrity of the mitochondrial membrane and ATP level [64]. Briefly, about 500 mg of tissue sample was cut into small pieces and homogenized with a Teflon pestle in 1 mL Isolation Medium (IM; 10 mM Tris, 250 mM sucrose, 0.5 mM Na_2_EDTA, and 1 g/L free fatty acid BSA, pH 7.4, 4 °C). The obtained homogenate was centrifuged at 1000× *g* for 10 min at 4 °C, and the resultant supernatant was transferred to a new microcentrifuge tube for high-speed centrifugation (15,000× *g*) for 20 min at 4 °C. The supernatant containing the cytosolic protein fractions was collected, and the pellet of intact mitochondrial fractions were resuspended in a 1 mL IM (BSA-free) followed by incubation on ice for 30 min. The concentration of protein in both fractions was determined by the Bradford method.

### 2.6. Determination of Mitochondrial ATP Levels

The adenosine 5-triphosphate (ATP) concentration in the isolated mitochondrial fraction from RVL samples was analyzed by a luciferase-based assay kit (Molecular Probes, A22066; Invitrogen, Madrid, Spain), according to the manufacturer’s instructions. The luminescence was recorded by a luminometer (TECAN-Infinite 200 PR) at a wavelength of 560 nm. The detection sensitivity of this assay is higher than 0.1 pM/mg protein of ATP. Experiments were performed at 0.5 mg/mL protein concentration. Mitochondrial ATP levels were expressed as nanomoles of ATP per milligram of protein.

### 2.7. Measurement of Cytosolic and Mitochondrial Nitrite Levels

This methodology entails the employment of the colorimetric Griess diazotization reaction to spectrophotometrically discern the presence of nitrite generated through the spontaneous oxidation of nitric oxide (NO) under physiological conditions [65]. Nitrite levels were quantified in the cytosolic and mitochondrial fractions from RVL samples using Griess Reagent Kit (G-7921; Molecular Probes, Life Technologies, Madrid, Spain), as described in the manufacturer’s instructions. The minimum detection limit of this method is 1.0 µM nitrite. Nitrite levels were expressed as micromoles per milligram of protein.

### 2.8. Measurement of Cytosolic and Mitochondrial SOD Activity

The Mn and Cu/Zn superoxide dismutase activity (SOD) in the cytosolic and mitochondrial fractions from RVL samples was quantified using a SOD Assay Kit (CS0009, Sigma-Aldrich, Madrid, Spain), following the manufacturer’s instructions. The kit’s lowest limit of detection is 0.3 units/mL SOD. SOD units were normalized to milligram of total protein concentration, and the SOD activity was expressed as a percentage of the inhibition rate.

### 2.9. Histological Analysis

The OCT-embedded RVL samples were cross-sectioned using a LEICA CM1510S Cryostat microtome at −20 °C with a thickness of 12 or 15 μm. The frozen cryosections were assessed for tissue structure detail, fiber oxidative capacity, and ectopic lipid deposition. The general histological pattern was assessed using a standard Hematoxylin and Eosin (H&E) staining protocol. Additional cryosections were assessed for the oxidative capacity of muscle fibers using nicotinamide adenine dinucleotide tetrazolium reductase (NADH-TR) histochemical staining, as described previously [66]. The NADH-TR reaction permits the differentiation of fibers according to their staining intensity. The muscular lipids content was detected in RVL muscle by ORO soluble dye (O-0625, Sigma-Aldrich, Spain) using standard protocols on cryosections fixed in 4% paraformaldehyde.

Following all staining, images of tissue cross-sections were captured at 20× objective using a light microscope (Olympus, Germany) equipped with a digital camera system (Carl Zeiss camera, model Axiocam ERC 5s. 222 Göttingen, Germany). Fiber cross-sectional area (CSA) was measured from cross-sections stained with H&E, according to the method described by Ceglia et al. [67]. Briefly, we outlined the circumference of each single muscle fiber using ImageJ 1.33 software (National Institutes of Health, Bethesda, MD, USA) and only those fibers with distinctly transverse sections and clear boundaries were included in the measurements. Fibers with marked signs of distortion or folding and those presenting oblique sections that appeared elongated in shape were not considered in the measurements [67]. The number of oxidative fibers were scored from cross-sections stained with NADH-TR and expressed as a percentage of the total number of fibers. The densitometrical quantification of the NADH-TR- and ORO-stained sections was assessed by mean pixel density using ImageJ 1.33 software (National Institutes of Health, Bethesda, MD, USA) from grayscale images of the same threshold, and data were defined as a percentage of stained area to a total cross-sectional area of the tissue. An average of five transverse sections per animal in each experimental group were used for the morphometric measurements. Multiple fields of view (20×) within each section were selected for the average CSA measurement and fiber counting. Each field area is 323 cm^2^ and corresponds to approximately 130 to 160 fibers. Five contagious fields (20×) of view within each section that was free of artifacts were analyzed for NADH activity and lipid content.

### 2.10. Total Lipid Extraction and Quantification

Lipids were isolated from about 100 mg of RVL samples in Chloroform/Methanol (2:1 *v*/*v*) and homogenized with a Dounce homogenizer at 4 °C. Homogenates were vortexed for 2 min, filtered on a 40 µm-membrane, and then centrifuged at 1000× *g* for 10 min at 4 °C. The lower lipid-containing phase was collected, and a 1/3 volume of 0.88% KCl was added. Then, the mixture was vortexed for 30 s and centrifuged at 1000× *g* for 10 min at 4 °C. The final lower organic phase was collected, dried at 90 °C, dissolved in 1 µL of Chloroform, and then stored at −20 °C.

Lipid quantification was determined by the Sulfuric-Phosphoric-Vanillin (SPV) colorimetric method using olive oil as a lipid amount standard, as described previously [68]. The absorbance was measured at 540 nm, and the lipid amount was expressed as a microgram per 100 milligrams of tissue.

### 2.11. Statistical Analysis

All experiments were performed in three independent experiments and repeated at least in duplicate in each rat to ensure reliability. All results were expressed as means ± standard deviation (SD) values. Comparisons between experimental groups were analyzed using one-way ANOVA followed by the post hoc Tukey’s Test. Statistically significant differences between groups were established if the *p*-value < 0.05. SPSS version 22 for Windows (SPSS, Michigan, IL, USA) was used for statistical data analyses.

## 3. Results

### 3.1. Melatonin Reduces Fluid Intake and Improves the Red Vastus Lateralis Muscle Mass of ZDF Rats

As depicted in Table 1, the daily feed intake per individual rat over the experiment period was markedly higher in the C-ZDF group than in the C-ZL group (*p* < 0.01). The feed conversion ratio (FCR), which defines gram feed per gram BW gain, was significantly increased in the C-ZDF group compared to the C-ZL group (*p* < 0.01; Table 1). However, there were no significant differences between the treated groups and their respective control groups for both endpoints (*p* > 0.05; Table 1).

We recorded a significant elevation of the daily fluid ingestion throughout the experiment in the C-ZDF rats compared with the C-ZL rats (*p* < 0.01; Table 1). Melatonin treatment resulted in a significant reduction in daily fluid intake in the M-ZDF group but not in the M-ZL group, as compared to their control counterparts (*p* < 0.01 and *p* > 0.05, respectively; Table 1).

The RVL weight was significantly lower in the C-ZDF than in the C-ZL group (*p* < 0.05; Table 1), and the extent of this difference was greater when the values were normalized to 100 g of BW (*p* < 0.01; Table 1). Melatonin treatment was found to increase the RVL weight in both the M-ZDF and M-ZL groups compared to their respective control (*p* < 0.05 and *p* < 0.05, respectively; Table 1). Likewise, the RVL weigh normalized to 100 g BW yielded significantly higher values in the M-ZDF and, interestingly, even in the M-ZL groups compared to their control counterparts (*p* < 0.01 and *p* < 0.01, respectively; Table 1).

### 3.2. Melatonin Improves Mitochondrial Fission/Fusion Balance and Autophagy in the Red Vastus Lateralis Muscle of ZDF Rats

Mitochondrial dysfunction has recently emerged as a hallmark of obesity and a key pathophysiological factor in different metabolic tissues, including SKM. Because mitochondrial fission/fusion and autophagy are two critical dynamic processes for maintaining functional mitochondria, we were interested in understanding how obesity affects mitochondrial fission/fusion balance and autophagy in the RVL muscle and whether melatonin treatment had any beneficial effects. 

Western blot analysis demonstrated that the Fis1 and DRP1 expressions were significantly lower in the C-ZDF group (1.69 ± 0.04 AU and 1.86 ± 0.03 AU, respectively) than in the C-ZL group (2.25 ± 0.04 AU; *p* < 0.01 and 2.28 ± 0.08 AU; *p* < 0.05, respectively; Figure 1A,B). After melatonin treatment, the protein expression of Fis1 was significantly increased in both ZDF (2.05 ± 0.03 AU) and ZL (2.46 ± 0.05 AU) groups compared with their non-treated counterparts (*p* < 0.05 and *p* < 0.05, respectively; Figure 1A), while that of DRP1 was significantly up-regulated only in the ZDF group (2.10 ± 0.11 AU) but not in the ZL group (2.30 ± 0.05 AU), as compared to their control counterparts (*p* < 0.05 and *p >* 0.05, respectively; Figure 1B). On the other hand, OPA1 and Mfn2 exhibited significantly higher protein level in the C-ZDF group (2.85 ± 0.10 AU and 4.76 ± 0.18 AU, respectively) that in the C-ZL group (2.16 ± 0.11 AU; *p* < 0.05 and 3.28 ± 0.20 AU; *p* < 0.05, respectively; Figure 1C,D), and melatonin treatment was found to significantly downregulate their protein levels in both ZDF (2.35 ± 0.09 AU and 3.64 ± 0.08 AU, respectively) and ZL (1.85 ± 0.01 AU and 2.71 ± 0.18 AU, respectively) groups, as compared with their corresponding C-ZDF (*p* < 0.05 and *p* < 0.01, respectively) and C-ZL (*p* < 0.05 and *p* < 0.05, respectively) groups (Figure 1C,D). The ratio of DRP1 to Mfn2 expression was significantly lower in the C-ZDF group than the C-ZL group (2-fold; *p* < 0.05), and treatment with melatonin resulted in a significantly higher ratio in both the ZDF (1.8-fold) and ZL (1.3-fold) groups, as compared to their control counterparts (*p* < 0.05 and *p* < 0.05; Figure 1E).

While both excessive and defective autophagy have been proposed as a contributing mechanism to the onset of many diseases, the role of autophagy in obesity-associated SKM damage has rarely been studied. To investigate whether and how autophagy in SKM relates to obesity and if treatment with melatonin could revert any possible alteration, we measured the expression levels of p62/SQSTM1 and total LC3b as autophagy effectors in the RVL muscle from obese ZDF rats using Western blot. As depicted in Figure 2A,B, the protein levels of p62/SQSTM1 and total LC3b were significantly lower in the C-ZDF group (0.65 ± 0.04 AU and 1.07 ± 0.01 AU, respectively) than in the C-ZL group (2.10 ± 0.12 AU; *p* < 0.01 and 2.24 ± 0.10 AU; *p* < 0.01, respectively), and melatonin treatment was found to significantly up-regulate the expression of p62/SQSTM1 and total LC3b in the ZDF group (1.21 ± 0.09 AU and 1.88 ± 0.04 AU, respectively) but not in the ZL group (2.07 ± 0.11 AU and 2.32 ± 0.01 AU, respectively), as compared with their respective C-ZDF (*p* < 0.01 and *p* < 0.01, respectively) and C-ZL (*p* > 0.05 and *p* > 0.05, respectively) groups.

To better understand the effect of melatonin on the autophagy process in the RVL muscle under obese and diabetic states, we further evaluated the LC3bI (inactive cytosolic form) and LC3bII (active membrane-bound form) levels by Western blot analysis. As shown in Figure 2C,D, Western blotting indicated significantly lower LC3bI protein level in the C-ZDF group (0.49 ± 0.03 AU) than in the C-ZL group (0.71 ± 0.02 AU; *p* < 0.05), while the protein level of LC3bII exhibited no significant difference between both groups (0.48 ± 0.02 AU and 0.47 ± 0.02 AU; *p* > 0.05, respectively). We also considered the ratio LC3bII to LC3bI as a marker of autophagy and found that it was significantly increased in C-ZDF group compared to the C-ZL group (1.5-fold *p* < 0.05; Figure 2E). After melatonin treatment, the protein level of LC3bI was significantly increased in the ZDF group (0.78 ± 0.01 AU) but not in the ZL group (0.74 ± 0.03 AU), whereas that of LC3bII was attenuated in both ZDF (0.37 ± 0.01 AU) and ZL (0.31 ± 0.01 AU) groups, as compared with their respective C-ZDF (*p* < 0.05 and *p* < 0.05, respectively) and C-ZL (*p* > 0.05 and *p* < 0.01, respectively) groups (Figure 2C,D). These changes were reflected in a significant decrease in the ratio of LC3bII to LC3bI in both C-ZDF (2.1-fold) and C-ZL (1.6-fold) groups in comparison with their non-treated counterparts (*p* < 0.01 and *p* < 0.01, respectively; Figure 2E).

The expression of SIRT1 as a regulator of the autophagy pathway in the RVL muscle was also considered. Western blot analysis showed significantly lower SIRT1 expression in the C-ZDF group (0.67± 0.01 AU) than in the C-ZL group (1.21 ± 0.03 AU; *p* < 0.01), and that melatonin treatment significantly up-regulated its expression in both ZDF (0.83 ± 0.04 AU) and ZL (1.35 ± 0.05 AU) groups compared with their respective control (*p* < 0.05 and *p* < 0.05, respectively; Figure 2F).

### 3.3. Melatonin Improves ATP Production and Prevents Nitro-Oxydative Stress in the Red Vastus Lateralis Muscle of ZDF Rats

ATP depletion and oxidative stress are two major triggers of mitochondrial dysfunction. Accordingly, we were interested in investigating whether alterations in the mitochondrial fission/fusion balance and autophagy process in the RVL muscle of non-treated ZDF rats coincided with mitochondrial ATP level depletion and oxidative stress and whether treatment with melatonin exerted any protective effects against such deleterious situations.

As shown in Figure 3, the mitochondrial ATP level was significantly lower in the C-ZDF group (0.37 ± 0.04 nmol/mg protein) compared with the C-ZL group (0.59 ± 0.02 nmol/mg protein; *p* < 0.01), and it was significantly increased after melatonin treatment in the ZDF group (0.58 ± 0.03 nmol/mg protein) but not in the ZL group (0.58 ± 0.04 nmol/mg protein) compared with their respective control groups (*p* < 0.01 and *p* > 0.05, respectively).

We examined oxidative stress by measuring the nitrite level and the enzymatic activity of the antioxidant Mn- and CuZn-SOD in both cytosolic and mitochondrial fractions isolated from the RVL muscle. The C-ZDF group exhibited significantly higher cytosolic and mitochondrial nitrite levels in the RVL tissue (4.6 ± 0.1 μmol/mg and 41.3 ± 0.18 μmol/mg, respectively) than the C-ZL group (3.5 ± 0.4 μmol/mg; *p* < 0.05; Figure 4A and 27.7 ± 1.3 μmol/mg; *p* < 0.01; Figure 4B, respectively). Melatonin treatment significantly lowered cytosolic and mitochondrial nitrite levels in both the ZDF (2.5 ± 0.2 μmol/mg and 26.5 ± 0.5 μmol/mg) and ZL (1.8 ± 0.2 μmol/mg and 22.2 ± 1.1 μmol/mg) groups, as compared to their respective C-ZDF (*p* < 0.01; Figure 4A and *p* < 0.01; Figure 4B, respectively) and C-ZL (*p* < 0.01; Figure 4A and *p* < 0.05; Figure 4B, respectively) groups. Furthermore, the cytosolic and mitochondrial SOD activity was significantly lower in RVL tissue of the C-ZDF group (55.0 ± 1.0% and 60.0 ± 3.0%, respectively) than in the C-ZL group (76.9 ± 0.8%; *p* < 0.01; Figure 5A and 91.6 ± 4.0% *p* < 0.01; Figure 5B). Treatment with melatonin significantly increased the cytosolic and mitochondrial SOD activity in both the ZDF (62.2 ± 1.9% and 75.9 ± 2.0%, respectively) and ZL (87.6 ± 1.8% and 100.0% ± 2.5, respectively) groups, as compared to their respective C-ZDF (*p* < 0.01; Figure 5A and *p* < 0.01; Figure 5B, respectively) and C-ZL (*p* < 0.01; Figure 5A and *p* < 0.05; Figure 5B, respectively) groups.

### 3.4. Melatonin Improves MyH Isoforms Distribution and Fiber Oxydative Capacity in the Red Vastus Lateralis Muscle of ZDF Rats

Mitochondria dysfunction is considered to play a causal role in obesity-induced muscle fiber remodeling [5]. To explore whether the substantial effects of melatonin on mitochondria dynamics were reflected in enhanced fiber type composition, fiber type identity based on the content of different MyH isoforms and histochemical NADH enzyme activity was performed.

We investigated the four fiber types in mammalian SKM (type I, IIa, IIx, and IIb), according to MyH isoform Western blotting: MyH for type I (slow/oxidative), MyH for IIa (fast/oxidative), MyH for IIx (fast/glycolytic), and MyH for IIb (fast/glycolytic). As depicted in Figure 6, whereas MyH1 expression (0.43 ± 0.06 AU) remained unchanged in the C-ZDF group, the expression of MyH7 (0.28 ± 0.05 AU) and MyH2 (0.71 ± 0.05 AU) reduced and that of MyH4 (0.87 ± 0.04 AU) was significantly increased when compared to the C-ZL group (0.44 ± 0.08 AU, 0.97 ± 0.07 AU, 1.24 ± 0.06 AU, 0.14 ± 0.05 AU, respectively; *p* > 0.05, *p* < 0.01, *p* < 0.01, *p* < 0.01, respectively). Melatonin treatment significantly counteracted the obesity-induced changes for MyH7 (0.85 ± 0.05 AU), MyH2 (1.10 ± 0.07 AU), and MyH4 (0.21 ± 0.05 AU) expression in the ZDF group while conversely had no effects for MyH1 expression (0.33 ± 0.05 AU), as compared to the respective control group (*p* < 0.01, *p* < 0.01, and *p* < 0.01, respectively; Figure 6). Interestingly, the expressed amount of MyH7, MyH2, and MyH4 in the M-ZDF group was returned to those of the C-ZL group. In the M-ZL group, MyH7 (1.12 ± 0.07 AU) and MyH2 (1.37 ± 0.06 AU) protein expression tended to be increased, while that of MyH4 (0.13 ± 0.05 AU) seemed to be slightly decreased, as compared to the respective control group, albeit without statistical significance (*p* > 0.05, *p* > 0.05 and *p* > 0.05, respectively; Figure 6). In contrast, the MyH1 level remained unchanged in the M-ZL group (0.36 ± 0.47 AU) compared to the control counterpart (*p* > 0.05; Figure 6).

The concept of muscle fiber type relates to its metabolic properties [5]. To further evaluate the fiber type identity in terms of oxidative capacity, we analyzed NADH-TR mitochondrial enzyme activity on RVL cross-sections. The enzymatic activity of NADH is commonly used as a marker of mitochondrial content and oxidative capacity of muscle fibers [69]. The oxidative capacity of fibers was subjectively evaluated according to the blue staining intensity and scored as oxidative fibers when moderate to strong staining was easily detectable or as non-oxidative fibers when there was no apparent staining within the fiber. As shown in Figure 7A, the RVL muscle from the C-ZL group comprises a mix of oxidative fibers and non-oxidative ones. However, the RVL muscle from the C-ZDF group appears to consist predominately of non-oxidative fibers, whereas few oxidative fibers are detected (Figure 7A). The RVL muscle from the M-ZDF group presented a similar NADH-TR staining pattern to the C-ZL group (Figure 7A). Quantitatively, the frequency of oxidative fibers was markedly reduced in the RVL muscle of the C-ZDF group (36.7 ± 1.9%) in comparison to the C-ZL group (54.5 ± 1.9%; *p* < 0.01; *p* < 0.01; Figure 7B), but distinctly increased in the M-ZDF group (34.9 ± 2.0%) relative to the C-ZDF group (*p* < 0.01; Figure 7B). However, there was no significant difference in the frequency of oxidative fibers between the M-ZL (42.5 ± 2.3%) and C-ZL groups (*p* > 0.05; Figure 7B). The NADH-TR intensity of muscle fibers was further examined densitometrically and found to be significantly lower in the C-ZDF group (21.31 ± 2.9%) in comparison with the C-ZL group (38.6 ± 2.5%; *p* < 0.01; Figure 7B). Melatonin treatment significantly increased the NADH-TR intensity in the M-ZDF group (40.0 ± 2.4%), as compared to the C-ZDF group (*p* < 0.01; Figure 7B). However, there was no significant difference in the NADH-TR intensity between the M-ZL group (40.9 ± 2.6%) and the C-ZL group (*p* > 0.05; Figure 7C).

Higher oxidative capacity within muscle fibers reflects enhanced capacity for lipid oxidation [5]. To confirm the changes in the oxidative capacity of fibers, all RVL muscles underwent qualitative and quantitative measurements of lipid content. As expected, visualization by ORO staining demonstrated striking lipid intramyocellular accumulation (IMCL) in the RVL muscle of the C-ZDF groups compared to the C-ZL group, whereas a weak IMCL positivity was noted in the M-ZDF group (Figure 8A). Quantitation of ORO staining intensity revealed significantly higher pixel intensity in the C-ZDF group (28.16 ± 1.3%) than in the C-ZL group (8.7 ± 2.0%; *p* < 0.01; Figure 8B). After melatonin treatment, the pixel intensity of ORO stain was significantly lower in the M-ZDF group (9.6 ± 1.1%) than the control counterparts (*p* < 0.05; Figure 8B); however, a non-significant tendency towards lower ORO intensity was noted in the M-ZL group (6.8 ± 1.7%) compared to the respective control group (*p* > 0.05; Figure 8B). These results were paralleled by a significant increase in total lipid content in the RVL muscle extracts from the C-ZDF group (49.0 ± 3.3 µg/100 mg tissue), as compared to the C-ZL group (37.5 ± 1.4 µg/100 mg tissue; *p* < 0.01; Figure 8C). The extent of this excessive total lipid content was significantly attenuated by melatonin treatment in the ZDF group (38.2 ± 3.3 µg/100 mg tissue) compared to the control counterpart (*p* < 0.01; Figure 8C) and normalized up to C-ZL level. Interestingly, melatonin treatment was found even to reduce the total lipid content in the RVL muscle of the ZL group (29.6 ± 3.8 µg/100 mg tissue) compared to the respective control (*p* < 0.05; Figure 8C).

To assess any eventual structural changes in the RVL muscle, we carried out H&E staining in cross-sections of the RVL muscle. As shown in Figure 9A, the cross-sections of the C-ZDF group displayed no particular sign of tissue damage mainly characterized by the separation of the fiber’s fascicles and presence of an edema with moderate fiber necrosis. However, the mean CSA of fibers showed a slight reduction in the C-ZDF group (21.6 ± 1.6 μm^2^ × 100) in comparison to the C-ZL group (25.6 ± 1.3 μm^2^ × 100; *p* < 0.05; Figure 9B). Interestingly, the mean CSA was significantly increased in both the M-ZDF (27.2 ± 1.1 μm^2^ × 100) and M-ZL (30.7 ± 1.7 μm^2^ × 100) groups, as compared to their control counterparts (*p* < 0.05 and *p* < 0.05, respectively; Figure 9B).

## 4. Discussion

We showed that melatonin treatment in obese and diabetic ZDF rats significantly prevents structural muscle damage. Specifically, melatonin treatment improved fiber atrophy and restored the muscle oxidative metabolic phenotype blunted by obesity, thereby leading to lower muscular lipid deposition, as evidenced by quantitative and qualitative analysis. At the molecular level, we revealed that melatonin treatment could improve muscle antioxidant capacity and ATP level by improving the mitochondrial fusion/fission balance and autophagy. These findings showed for the first time the role of melatonin in regulating SKM mitochondrial dynamic under obese and diabetic states, which may provide novel insights into potential therapeutic targets for SKM abnormality in obesity and T2DM in which mitochondrial dysfunction occurs.

The present results showed a substantial elevation of the daily fluid ingestion over twelve weeks in C-ZDF rats compared with C-ZL rats consistent with our previous finding in the same animal strain [70]. Increased fluid consumption is a well-known symptom of T2DM in human and animal models, which is the direct consequence of insulin deficiency. In line with our previous study [70], an extended treatment period (12 weeks) could subsequently reduce water intake without affecting feed intake. This substantial effect of melatonin on water intake could be attributable to better renal function. Consistent with this suggestion, our previous studies in obese and diabetic ZDF rats have demonstrated that melatonin treatment improved renal function and morphology by preventing renal endoplasmic reticulum stress and mitochondrial dynamic imbalance [71,72].

At the molecular level, we showed that obese and diabetic ZDF rats display prominent mitochondrial dynamic alterations in the RVL muscular tissue as evidenced by lower expression of fission markers (Fis1 and DRP1) and higher fusion markers (Mfn2 and OPA1) expression when compared to their lean littermates, indicating the disturbance of the balance between mitochondrial fission and fusion. This suggestion was supported by the decreased DRP1/Mfn2 ratio of about 58% in obese and diabetic ZDF rats compared to their lean littermates. Any disbalance in fusion and fission machinery may be promptly reflected by ultrastructural disorganization [73]. Although these results most strongly indicate the imbalance of mitochondrial fission/fusion in the RVL muscle of obese and diabetic ZDF rats, further experimental validation using an electronic microscope approach is needed to explore mitochondrial networking and further support this observation. Results obtained in this work are supported by previous observations showing that the SKM of individuals with obesity and T2DM displays lower expression of Fis1 and DRP1 and higher Mfn1/2 expression relative to healthy subjects [41,74,75]. There has recently been growing evidence that altered fission or fusion SKM might induce pathways that effect muscle mass and metabolic flexibility, leading to muscle atrophy and influencing whole-body homeostasis [11,44]. Considering that aberrant mitochondrial fusion/fission has been highlighted in various pathological conditions of SKM, studies that explore potential pharmacological agents targeting mitochondrial dynamic in SKM under obese conditions have garnered much interest recently [42]. Here, we showed that melatonin treatment prevents the mitochondrial fission/fusion imbalance by decreasing OPA1 and Mfn2 expression and enhancing the expression of Fis1 and DRP1. These results suggest that the mitochondrial fission/fusion process may represent a significant target for melatonin in augmenting SKM mitochondrial functions, hence preventing obesity-related SKM metabolic disruptions. The capacity of melatonin to regulate mitochondrial dynamics aligns with prior findings in other insulin-sensitive tissues subjected to diverse pathological conditions [58,72,76,77,78,79]. Additionally, melatonin has been found to prevent differentiated SKM cell apoptosis and mitochondrial dysfunction by improving mitochondrial dynamic balance [61].

Autophagy is an essential catabolic process for the removal of damaged proteins and organelles by internalizing them in an autophagosome [80]. The present results showed a lower p62/SQSTM1 protein level and higher CL3bII/CL3bI ratio in the RVL muscle of obese and diabetic ZDF rats compared to their lean littermates. Even though there are multiple protein complexes involved in the autophagic flux, we selected key players that are essential for this process (p62 and CL3) to investigate the role of melatonin in this context. LC3b and p62 are the major proteins participating in autophagy, and it has been reported that changes in LC3b and p62 expression can be used as a reliable indicator for the constant evaluation of autophagy [81]. p62/SQSTM1 is an autophagosome adapter protein that binds ubiquitinated organelles [44]. LC3b is a cytosolic protein that is cleaved by protease ATG4 forming LC3bI that is then activated by ATG7, transferred to ATG10, and finally conjugated to a phosphatidylethanolamine of the phagophore membrane by the ATG5-12-16L complex forming the active LC3bII, which interacts with p62 conducting the targeted-organelle to the autophagosome [44]. When elongating the autophagosome membrane, LC3bI is converted to LC3bII [44]. The ratio of LC3bII to LC3bI is thus used to determine whether the autophagy pathway is activated. The present results are consistent with other studies showing increased LC3bII/I ratio in the SKM of diet-induced obesity models [82,83]. In humans, the autophagic/lysosomal pathway in the SKM has been found to be impaired under obese states along with impaired ubiquitin-proteasome system [84]. In addition, some studies have also shown that altered autophagy in SKM resulted in increased ROS/NRS generation, thus, leading to cell apoptosis and muscle atrophy and dysfunction [85].

In the present study, the SIRT1-mediated autophagy displayed a lower expression in the RVL muscle of obese and diabetic ZDF rats. SIRT1 has been recognized as one of the protective factors against pathological conditions linked to obesity by regulating autophagy and cell death [86]. Consistent with the present findings, decreased SIRT1 expression in the SKM has been shown to be associated with lipotoxicity, glucose intolerance, and insulin resistance [30,87]. The present findings showed that melatonin treatment restores the SIRT1 expression in obese diabetic rats by reestablishing the expression of autophagy markers, as evidenced by increased expression of p62/SQSTM1 and LC3bI and reduced LC3bII expression, which may contribute to the amelioration of mitochondrial dysfunction. In line with these findings, melatonin treatment has been shown to attenuate lipotoxicity in the liver of an obese experimental model by regulating mitochondrial autophagy and increasing SIRT1 expression [59]. Further findings from in vitro and in vivo models have shown that melatonin treatment improved mitochondrial autophagy in the SKM by inducing or inhibiting its activity, depending on the pathological condition, in concert with the improved morphological features of SKM [61,88,89,90]. Given that mitophagy is a dynamic process related to autophagy, further studies measuring this process in combination with autophagic flux and the ubiquitin-proteasome-associated degradation system will help to understand the mechanistic aspect regulating diabesity and to decipher the precise mechanism underlying melatonin.

ATP production is the main function of mitochondria to maintain the necessary cellular energy supply through coupled OXPHOS [91]. In the present study, we showed decreased ATP production in obese diabetic rats, suggesting alterations to the RVL mitochondrial function, and, interestingly, melatonin treatment restored its level to the normal range. Furthermore, the reduction in mitochondrial function observed in the RVL of ZDF rats may be due to an impaired OXPHOS and electron transport chain activities. In support of this idea, our previous studies have shown impaired OXPHOS and electron transport chain activities associated with depleted mitochondrial ATP levels and increased mitochondrial oxidative stress in the liver, kidney, and inguinal white and brown adipose tissues from the same obese model [72,76,78,79], that have been reversed by chronic melatonin treatment. It is plausible that melatonin may mitigate this impaired function by restoring ATP synthesis in mitochondria. Consistent with the current data, studies in animal models and humans have shown that obesity caused SKM mitochondrial dysfunction accompanied by decreased ATP production and OXPHOS enzyme activity, leading to an impaired ADP:O ratio and increased ROS production [39,92,93]. Also, elevated levels of ROS have been found to be correlated with increased levels of RNS in obesity and diabetes, causing cell damage and apoptosis [65,94]. Consistent with this idea, our present findings showed that the RVL of ZDF rats has higher levels of nitrites in the mitochondria, where they are generated, and in the cytosol as a consequence of mitochondrial dysfunction, and melatonin treatment was found to reduce nitrites levels in both lean and obese diabetic rats. In other pathological conditions, such as septic mice, melatonin could counteract cardiac and SKM mitochondrial dysfunction by ameliorating ATP production and OXPHOS coupling, minimizing ROS and RNS production [95,96]. Melatonin has antioxidant properties not only because of its chemical structure and its ability to reduce ROS/RNS but also because it increases the activity and expression of the endogenous antioxidant defenses such as SOD, catalase, and glutathione [57,97]. Increased meta-inflammation and oxidative stress are the classical hallmarks of obesity and are related to lower SOD enzyme expression and activity in several insulin-sensitive tissues, such as kidney, liver, adipose tissue, and SKM [72,78,79,98,99,100,101]. Consequently, we investigated the potential effects of melatonin treatment on the SOD activity in both mitochondrial and cytosolic RVL muscle fractions. To this end, we measured both isoforms of SOD, the activity of SOD Cu/Zn-SOD isoform (SOD1; ubiquitous enzyme), which has stable expression and the Mn-SOD isoform (SOD2), which is predominantly expressed in mitochondria and is *tightly* regulated [102]. As expected, obese ZDF rats exhibited lower SOD activity in both the mitochondrial and cytosolic RVL muscle fractions compared to their lean littermates and, interestingly, melatonin treatment partially recovers the deficient SOD activity, which could be indicative of improved antioxidant defenses. Further analysis of the antioxidant effects of melatonin in SKM is needed for a better understanding of the mechanisms underlying these beneficial effects.

The hallmarks of abnormal SKM in obesity are alteration of muscle fiber distribution, loss of muscle mass, and ectopic lipid deposition [5,103]. SKM has a heterogeneous composition with different fiber types with peculiar physiological contractile and metabolic properties [103]. Type I fibers (slow-switch oxidative fibers) predominantly utilize oxidative metabolism and exhibit greater resistance to fatigue, while type II fibers (fast-switch glycolytic fibers) rely on glycolytic metabolism and are more susceptible to fatigue [103]. SKM metabolism and performance depend upon muscle plasticity and, therefore, the relative proportion of the fiber types [103]. Western blotting of obese ZDF RVL samples showed a reduction in type I slow fiber markers MYH7 and type IIa fast fiber markers MYH2, which are characterized by the use of oxidative metabolism and, conversely, an increased proportion of type IIb fast fibers markers MYH4, which rely on glycolytic metabolism. These results seem compatible with the histochemical NADH activity results and thus support the idea that they might likely reflect the transition from oxidative to glycolytic phenotype. In line with our results in the ZDF model, some studies have reported that T2DM and obese subjects exhibited fewer type I muscle fibers and decreased oxidative enzyme activity, indicative of a blunted SKM oxidative capacity in obese subjects [104,105,106]. Importantly, there has been a reduced percentage of type I slow-switch oxidative fibers correlating negatively with higher insulin resistance in patients with type 2 diabetes [107]. The significant decrease in the proportion of MyH7 and MyH2 in obese ZDF rats could be explained by either degeneration of oxidative fibers or transformation into a glycolytic fibers type within the RVL muscle. We are in favor of the later hypothesis since it is supported by two important findings, showing that there was a significant increase in the proportion of MyH4 within the obese ZDF RVL muscle and no histological sign of muscle degeneration. On the other hand, the RVL muscle of the obese and diabetic ZDF group assumed a marked subnormality of RVL weights associated with slight fiber atrophy relative to the lean control group, indicative of muscle atrophy. Consistent with these results, the loss of muscle mass (muscle atrophy), termed sarcopenia, is one of the most established features of T2DM and obesity [108,109,110,111,112,113]. Because we did not note any histological signs of muscle damage in the obese ZDF rats, the observed muscle loss may be the underrepresentation of fiber atrophy rather than fiber degeneration due to a reduction in the MyH isoforms content. The Western blotting of MyH isoforms may support this idea since the total MyH isoforms were about 1.18-fold lower in the obese ZDF rats than in the lean control. However, recent studies showed that SKM plasticity under different conditions is not always accompanied by fiber switching and may vary the oxidative capacity and the fiber size without myosin heavy chain changes [114,115]. This fact could explain why even CSA of RVL is decreased in ZDF rats; the expression of slow-twitch fiber marker MyH7 was also decreased, showing that there is no direct relationship between fiber size and fiber type characterization. Consequently, further immunohistochemistry or immunofluorescence analyses are warranted to examine whether the changes observed are ascribable to a quantitative fiber type switching and not only to variations in mitochondrial activity or fiber size.

From a metabolic point of view, the reduction in the ratio of oxidative to type glycolytic fibers in the control ZDF rats led to a RVL muscle with a lower density of oxidative fibers and higher ectopic lipid deposition, as evidenced by the quantitative and qualitative analysis. Here, we demonstrated for the first time that melatonin treatment increases the proportion of the oxidative fibers in the RVL muscle of ZDF rats compared to ZDF rats without treatment, which tend to indicate an improved capacity of RVL muscle for oxidative capacity and oxidative utilization of free fatty acids. In support of this idea, our results further revealed a marked reduction in the quantitative total lipid content and qualitative IMCL deposition in the RVL muscle of ZDF rats, which strongly indicates the increased oxidation of lipids. Moreover, these results seem compatible with those of ATP level. Thus, these findings lend support to the idea of reversibility of RVL muscle oxidative capacity in response to melatonin treatment. Since SKM contributes to whole-body fatty acids utilization [5], the increased oxidative capacity in the RVL muscle of ZDF rats might explain the pronounced blood free fatty acids and triglycerides-lowering effects of melatonin treatment observed in our previous study [70]. On the other hand, these results might imply that the increased content of oxidative fibers in the RVL muscle of obese and diabetic rats through melatonin–substantiated by less muscular lipid deposition and increased muscle mass–improves the endurance capacity of the muscle since the muscular fatigue resistance is based on the activity and integrity of slow type I/oxidative fibers [116]. Also, numerous studies showed that melatonin increased muscle strength and endurance in sarcopenia and aging, suggesting similar effects on obesity [57]. Although further research on muscle endurance and strength will help to elucidate how obesity affects SKM in ZDF rats and whether melatonin could ameliorate these effects. Moreover, this study offers a new perspective on melatonin treatment, bringing to light its promotion of oxidative phenotype in SKM. The effects of melatonin treatment on muscle strength and endurance performances, that are known to be suboptimal or even compromised in obese and T2DM subjects, were not characterized since it was outside the scope of the present study. Nevertheless, further studies using the same animal strain would be required to delineate the role of melatonin in this context.

Therapeutic strategies involving targeting mitochondria and muscle fiber metabolism have been gaining increasing attention recently in the context of obesity [44,117]. Although many studies have demonstrated that exercise training promoted shifts between subtypes of type II fibers, it has been infeasible to induce changes between slow-switch type I oxidative and fast-switch type II glycolytic fibers through exercise, particularly in humans [47,118,119,120,121]. Hence, the present finding might be of particular relevance since melatonin was able to maintain an oxidative phenotype in the RVL muscle of ZDF rats similar to that of lean control, which requires further in-depth studies. Moreover, they suggested that the maintenance of a lean-like SKM metabolic phenotype by melatonin treatment might restore whole-body metabolism and enhance energy expenditure in muscles. Therefore, further studies are warranted to gain insight into the mechanistic link between the mitochondrial dynamics network and muscle metabolic remodeling and the precise molecular mechanisms through which melatonin regulates the metabolic remodeling of fibers. In addition, more studies are needed on skeletal muscles with different types of fiber composition, such as soleus (slow twitch) or extensor digitorum longus (fast twitch), and even taking into account the different fiber composition in different parts of one specific SKM (deep vs. superficial) to better understand the molecular regulation underlying SKM plasticity.

## 5. Conclusions

The results presented here reveal for the first time that chronic oral melatonin administration to obese Zücker rats induces remarkable muscle fiber type shift toward an oxidative phenotype and improves SKM mitochondrial dysfunction, probably by regulating mitochondrial fission/fusion balance and autophagy. The finding of this study supports the potential translation of melatonin use for treating human muscle metabolic inflexibility among obese and diabetic patients and brings us closer to developing a safe and effective therapeutic strategy for targeting mitochondrial dynamics and autophagy in SKM for the treatment of metabolic diseases, for which early attempts at effective drug treatment failed.

## Figures and Tables

**Figure 1 antioxidants-12-01499-f001:**
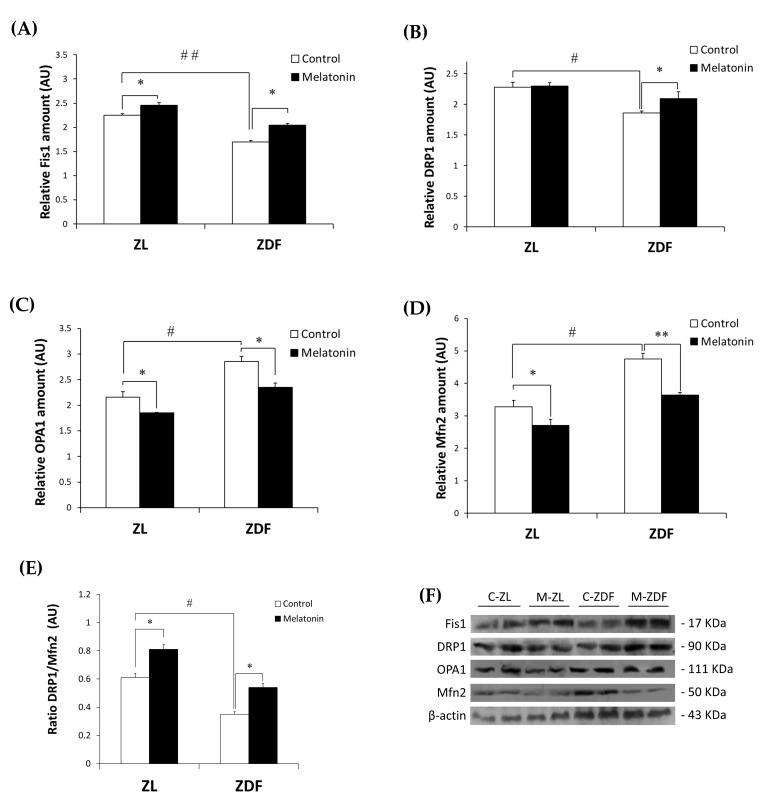
Effects of melatonin treatment on mitochondrial fission/fusion markers in the red vastus lateralis muscle of Zücker diabetic fatty rats and their lean littermates as measured by Western blot. (**A**–**D**) Densitometry quantification of fission 1 protein (Fis1), dynamin-related protein 1 (DRP1), optic atrophy protein type1 (OPA1), and mitofusin 2 (Mfn2) protein levels. (**E**) Ratio of DRP1 to Mfn2 expression. (**F**) Representative blots of Fis1, DRP1, OPA1, and Mfn2. C-ZL: control lean rats without melatonin; M-ZL: lean rats with melatonin; C-ZDF: control diabetic fatty rats without melatonin; M-ZDF: diabetic fatty rats with melatonin; ZL: Zücker lean rats; ZDF: Zücker diabetic fatty rat. All values are expressed as mean ± SD of three independent experiments performed in duplicate in each rat. The represented values were expressed relative to β-actin protein levels. ** *p* < 0.01; * *p* < 0.05; ## *p* < 0.01, # *p* < 0.05 (One-way ANOVA followed by Tukey post hoc test).

**Figure 2 antioxidants-12-01499-f002:**
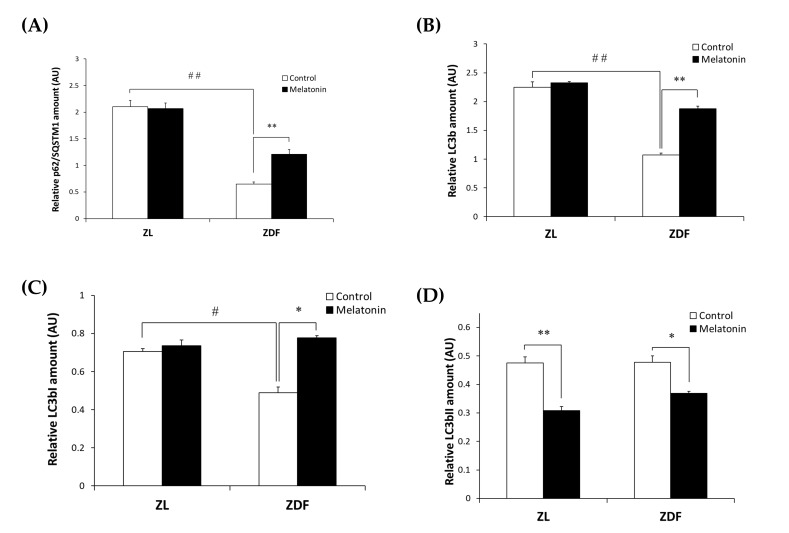
Effects of melatonin treatment on autophagy markers in the red vastus lateralis muscle of Zücker diabetic fatty rats and their lean littermates as measured by Western blot. (**A**–**D**,**F**) Densitometry quantification of sequestosome 1 (p62/SQSTM1), microtubule-associated protein 1 light chain 3b (LC3b), cytosolic inactive form of LC3b (LC3bI), membrane-bound active form of LC3b (LC3bII), and silent information regulator 1 (SIRT1) protein levels. (**E**) LC3bII/LC3bI ratio expression. (**G**) Representative blots of p62/SQSTM1, total LC3b, LC3bI, LC3bII, and SIRT1. C-ZL: control lean rats without melatonin; M-ZL: lean rats with melatonin; C-ZDF: control diabetic fatty rats without melatonin; M-ZDF: diabetic fatty rats with melatonin; ZL: Zücker lean rats; ZDF: Zücker diabetic fatty rat. All values are expressed as mean ± SD of three independent experiments performed in duplicate in each rat. The represented values were expressed relative to β-actin protein levels. ** *p* < 0.01; * *p* < 0.05; ## *p* < 0.01, # *p* < 0.05 (One-way ANOVA followed by Tukey post hoc test).

**Figure 3 antioxidants-12-01499-f003:**
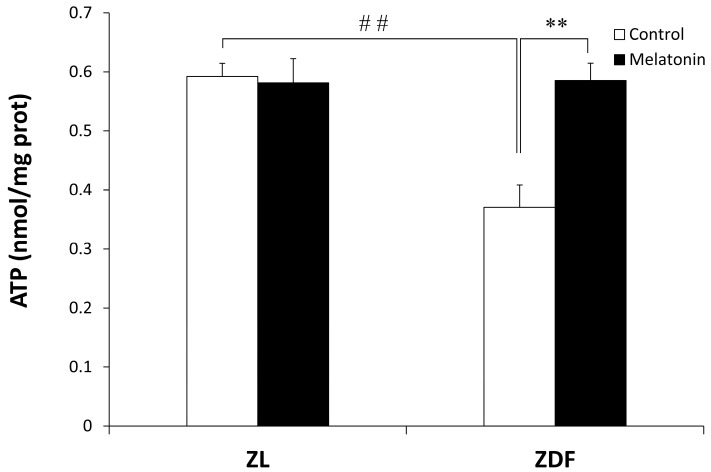
Effect of melatonin treatment on ATP levels in the mitochondrial fraction isolated from the red vastus lateralis muscle of Zücker diabetic fatty rats and their lean littermates. C-ZL: control lean rats without melatonin; M-ZL: lean rats with melatonin; C-ZDF: control diabetic fatty rats without melatonin; M-ZDF: diabetic fatty rats with melatonin; ZL: Zücker lean rats; ZDF: Zücker diabetic fatty rat. All values are expressed as mean ± SD of three independent experiments performed in triplicate in each rat. ** *p* < 0.01; ## *p* < 0.01 (One-way ANOVA followed by Tukey post hoc test).

**Figure 4 antioxidants-12-01499-f004:**
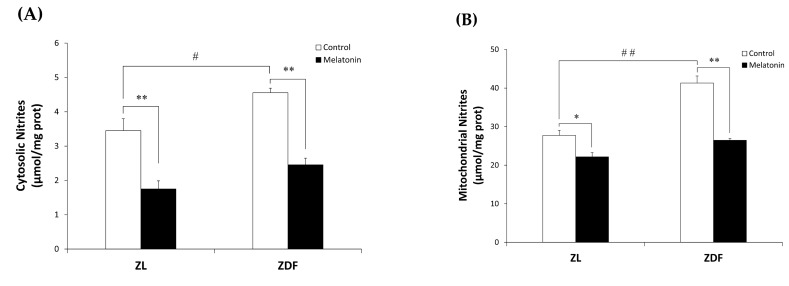
Effects of melatonin treatment on cytosolic and mitochondrial nitrite levels in the red vastus lateralis muscle of Zücker diabetic fatty rats and their lean littermates. (**A**,**B**) Nitrite levels in the cytosolic (**A**) and mitochondrial (**B**) muscle fractions. C-ZL: control lean rats without melatonin; M-ZL: lean rats with melatonin; C-ZDF: control diabetic fatty rats without melatonin; M-ZDF: diabetic fatty rats with melatonin; ZL: Zücker lean rats; ZDF: Zücker diabetic fatty rat. All values are expressed as mean ± SD of three independent experiments performed in duplicate in each rat. ** *p* < 0.01; * *p* < 0.05; ## *p* < 0.01, # *p* < 0.05 (One-way ANOVA followed by Tukey post hoc test).

**Figure 5 antioxidants-12-01499-f005:**
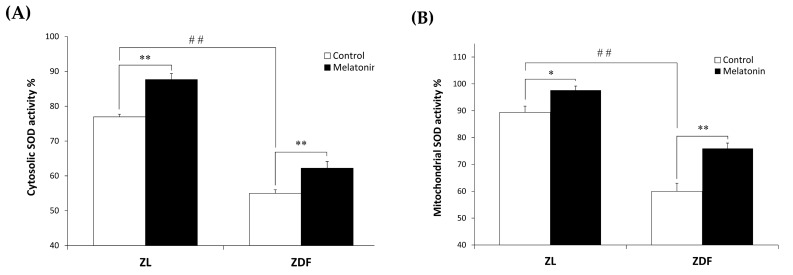
Effects of melatonin treatment on cytosolic and mitochondrial Mn- and CuZn- superoxide dismutase (SOD) activity in the red vastus lateralis muscle of Zücker diabetic fatty rats and their lean littermates. (**A**,**B**) Mn- and CuZn-SOD activity in the cytosolic (**A**) and mitochondrial (**B**) muscle fractions. C-ZL: control lean rats without melatonin; M-ZL: lean rats with melatonin; C-ZDF: control diabetic fatty rats without melatonin; M-ZDF: diabetic fatty rats with melatonin; ZL: Zücker lean rats; ZDF: Zücker diabetic fatty rat. All values are expressed as mean ± SD of three independent experiments performed in triplicate in each rat. ** *p* < 0.01; * *p* < 0.05; ## *p* < 0.01 (One-way ANOVA followed by Tukey post hoc test).

**Figure 6 antioxidants-12-01499-f006:**
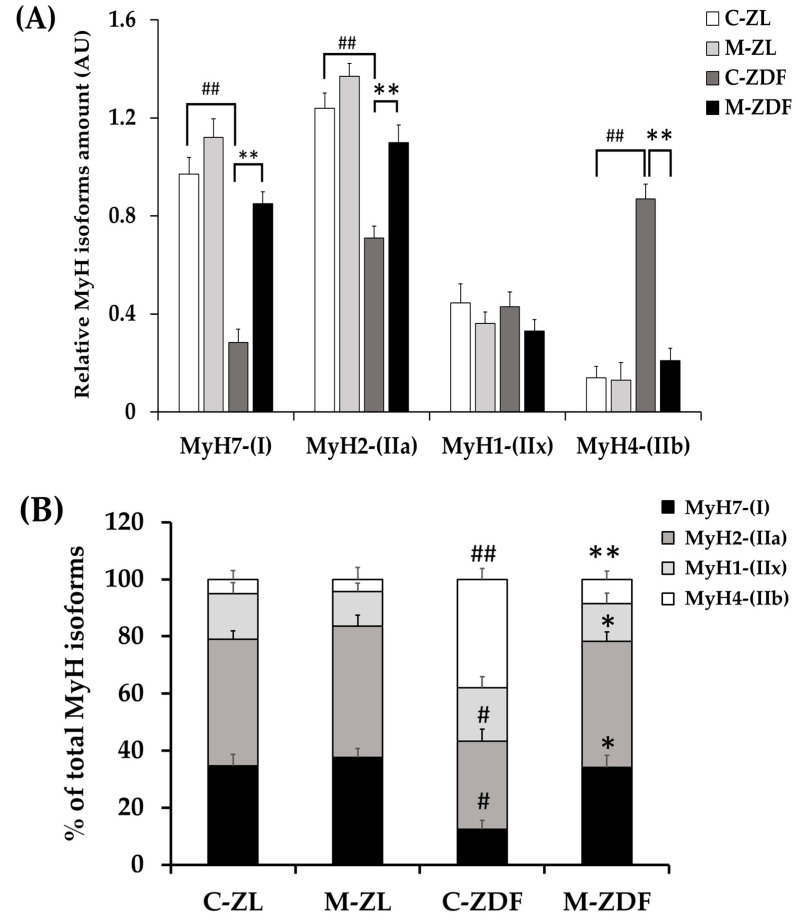
Effects of melatonin treatment on MyH isoforms distribution in the red vastus lateralis muscle of Zücker diabetic fatty rats and their lean littermates as measured by Western blot. (**A**) Densitometry quantification of Myosin heavy chain isoforms MyH7, MyH2, MyH4, and MyH1. (**B**) Relative expression of each MYH isoform was calculated as a percentage of the summed intensity of the four isoforms. (**C**) Representative blots of MyH7, MyH2, MyH4, and MyH1. C–ZL: control lean rats without melatonin; M–ZL: lean rats with melatonin; C–ZDF: control diabetic fatty rats without melatonin; M–ZDF: diabetic fatty rats with melatonin; ZL: Zücker lean rats; ZDF: Zücker diabetic fatty rat. All values are expressed as mean ± SD of three independent experiments performed in duplicate in each rat. The represented values were expressed relative to β-actin protein levels. ** *p* < 0.01; * *p* < 0.05; ## *p* < 0.01, # *p* < 0.05 (One-way ANOVA followed by Tukey post hoc test).

**Figure 7 antioxidants-12-01499-f007:**
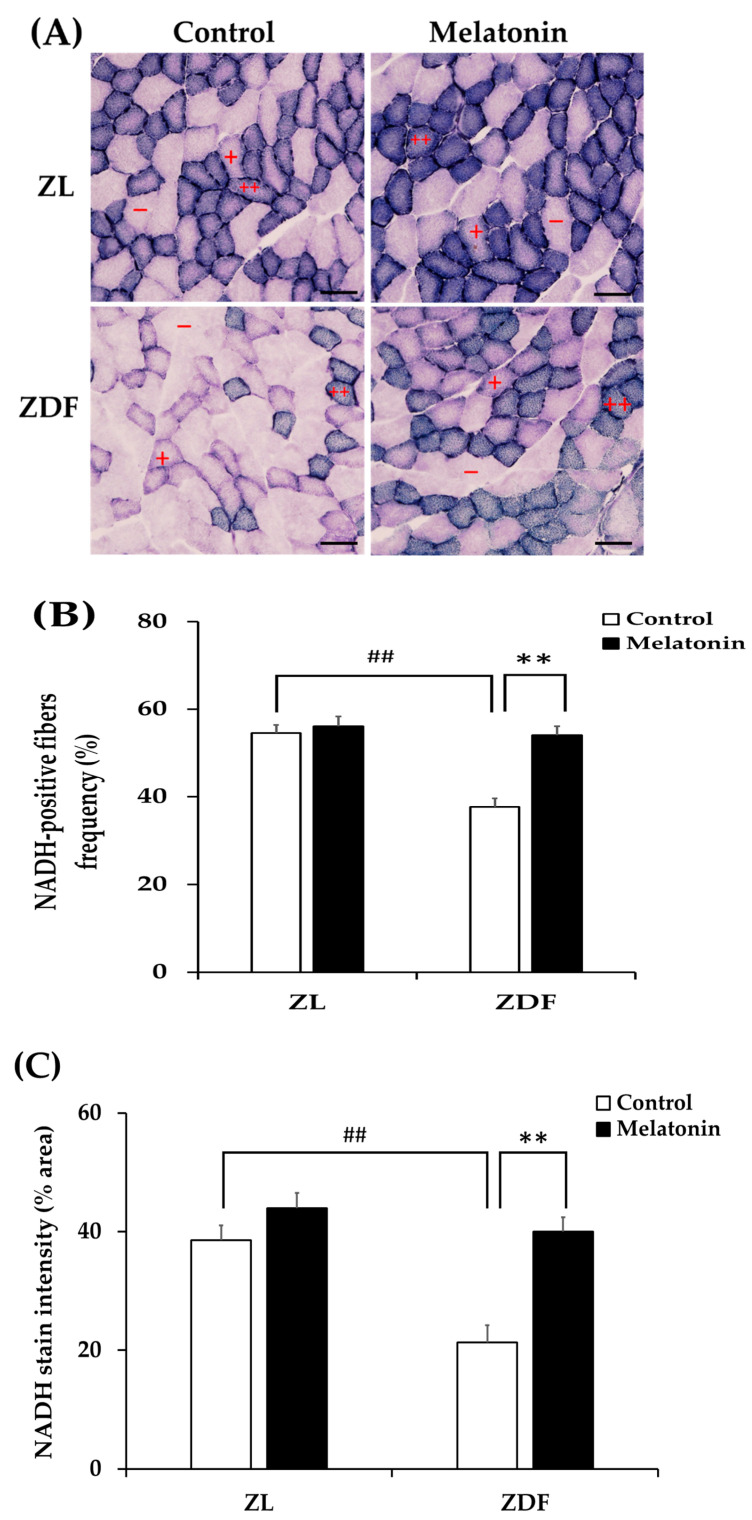
Effects of melatonin treatment on fiber oxidative capacity in the red vastus lateralis muscle of Zücker diabetic fatty rats and their lean littermates. (**A**) Representative cross-sections of histochemical enzymatic activity for nicotinamide adenine dinucleotide dehydrogenase tetrazolium reductase (NADH-TR). Oxidative fibers are dark blue (++) or intermediate blue (+) stained, while non-oxidative fibers are not stained (−). Few positively stained fibers were detected in C-ZDF rats, whereas an intense and diffuse positivity for NADH-TR was noted in M-ZDF and M-ZL rats. (**B**) Percentage of oxidative fibers to total fibers was calculated in multiple random fields. (**C**) NADH-TR stain intensity was assessed by mean pixel density and expressed as a percent area to the total area in multiple random fields. Fields of view were randomly selected from five transverse sections per animal. C-ZL: control lean rats without melatonin; M-ZL: lean rats with melatonin; C-ZDF: control diabetic fatty rats without melatonin; M-ZDF: diabetic fatty rats with melatonin; ZL: Zücker lean rats; ZDF: Zücker diabetic fatty rat. All values are expressed as mean ± SD. ## *p* < 0.01, ** *p* < 0.01. (One-way ANOVA followed by Tukey post hoc test). Original magnification at ×200. Scale bar: 100 μm.

**Figure 8 antioxidants-12-01499-f008:**
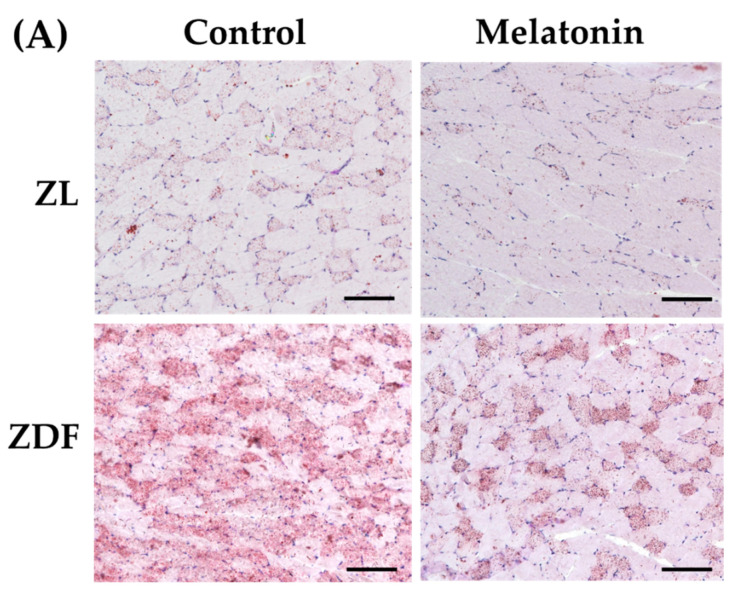
Effects of melatonin treatment on lipid accumulation in the red vastus lateralis muscle of Zücker diabetic fatty rats and their lean littermates. (**A**) Representative cross-sections of Oil-Red-O (ORO) staining showing excessive accumulation of intramyocellular neutral lipid viewed as aggregates of red spots was noted in C-ZDF rats compared to C-ZL rats, that was attenuated by melatonin treatment. (**B**) ORO intensity was assessed by mean pixel density and expressed as a percent to the total area in multiple of the selected five transverse sections per animal. (**C**) Quantification of muscle total lipid content. C-ZL: control lean rats without melatonin; M-ZL: lean rats with melatonin; C-ZDF: control diabetic fatty rats without melatonin; M-ZDF: diabetic fatty rats with melatonin; ZL: Zücker lean rats; ZDF: Zücker diabetic fatty rat. All values are expressed as mean ± SD. ** *p* < 0.01; * *p* < 0.05; ## *p* < 0.01. (One-way ANOVA followed by Tukey post hoc test). Original magnification at ×100. Scale bar: 200 μm.

**Figure 9 antioxidants-12-01499-f009:**
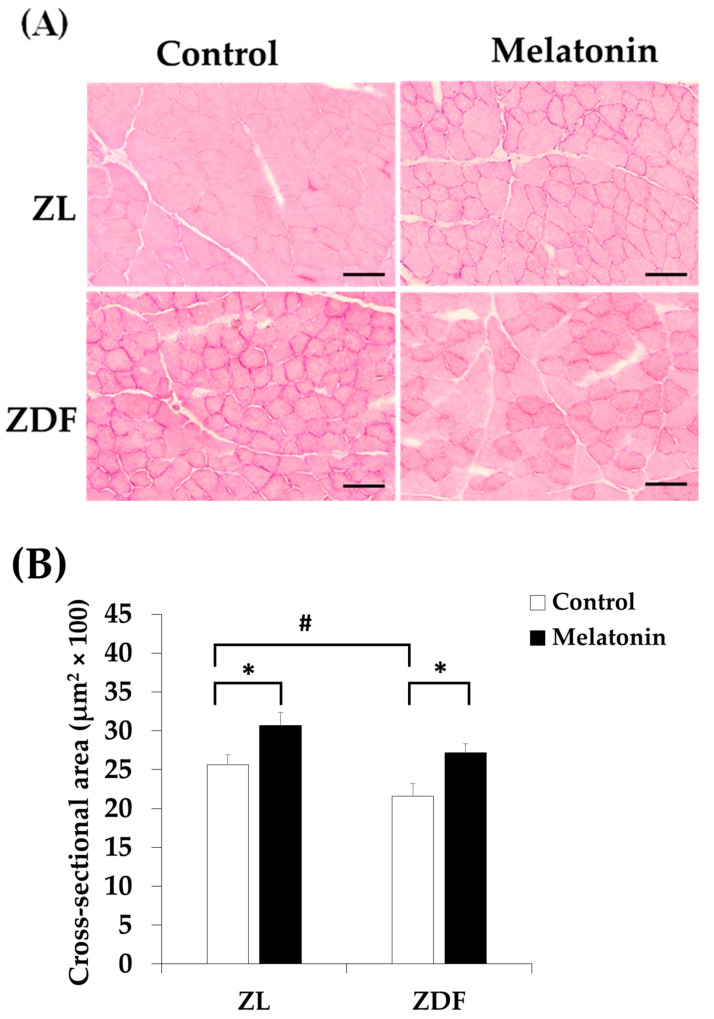
Effects of melatonin treatment on the histological structure of the red vastus lateralis muscle of Zücker diabetic fatty rats and their lean littermates. (**A**) Representative cross-sections of Hematoxylin and Eosin (H&E) showing no particular signs of tissue damage in C-ZDF rats. (**B**) Mean fiber cross-sectional area (CSA) averaged from 130 to 160 fibers per random field of five transverse sections per animal. C-ZL: control lean rats without melatonin; M-ZL: lean rats with melatonin; C-ZDF: control diabetic fatty rats without melatonin; M-ZDF: diabetic fatty rats with melatonin; ZL: Zücker lean rats; ZDF: Zücker diabetic fatty rat. All values are expressed as mean ± SD. * *p* < 0.05; # *p* < 0.05. (One-way ANOVA followed by Tukey post hoc test). Original magnification at ×200. Scale bar: 100 μm.

**Table 1 antioxidants-12-01499-t001:** Effects of melatonin treatment on feed intake, fluid intake, and red vastus lateralis muscle weight in Zücker diabetic fatty rats and their lean littermates.

Groups	Feed Intake (g/Day/Rat)	FCR (g Feed/g BW)	Fluid Intake (mL/Day/Rat)	RVL Weight (g)	Relative RVL Weight (g/100 g BW)
C-ZL	11.45 ± 0.40	4.02 ± 0.013	21.03 ± 0.58	2.50 ± 0.26	0.72 ± 0.03
M-ZL	11.29 ± 0.31	4.49 ± 0.24	20.78 ± 0.61	2.90 ± 0.09 ^‡^	0.98 ± 0.03 ^‡‡^
C-ZDF	26.02 ± 0.70 ^##^	6.60 ± 0.27 ^##^	93.24 ± 1.22 ^##^	1.96 ± 0.11 ^#^	0.44 ± 0.01 ^##^
M-ZDF	25.56 ± 0.38	7.07 ± 0.63	88.73 ± 1.54 **	2.31 ± 0.06 *	0.58 ± 0.01 **

C-ZL: control lean rats without melatonin; M-ZL: lean rats with melatonin; C-ZDF: control C-ZDF: diabetic fatty rats without melatonin; M-ZDF: diabetic fatty rats with melatonin; ZL: Zücker lean rats; ZDF: Zücker diabetic fatty rat. BW: body weight; FCR: feed conversion ratio (daily feed intake/BW gain); RVL: red vastus lateralis. Relative RVL weight was calculated as (total RVL weight/final BW) × 100. All values are expressed as mean ± SD (*n* = 8 rats/group). * *p* < 0.05, ** *p* < 0.01 M-ZDF vs. C-ZDF; # *p* < 0.05, ## *p* < 0.01 C-ZDF vs. C-ZL; ‡ *p* < 0.05, ‡‡ *p* < 0.01 M-ZL vs. C-ZL rats.

## Data Availability

Not applicable.

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
