# Peer review of "Melatonin Improves Skeletal Muscle Structure and Oxidative Phenotype by Regulating Mitochondrial Dynamics and Autophagy in Zücker Diabetic Fatty Rat"

_antioxidants, 2023, doi:10.3390/antiox12081499_

Round 1
Reviewer 1 Report (Previous Reviewer 3)
In their research article titled “Melatonin Improves Skeletal Muscle Structure and Oxidative Phenotype by Regulating Mitochondrial Dynamics and Mitophagy in Zücker Diabetic Fatty Rat”, Salagre and collaborators describe the effects of melatonin supplementation in lean and diabetic Zucker rats.
The work entails the analysis of the histological properties of skeletal muscle fibers together with expression of mitochondrial proteins of the red vastus lateralis muscle. The Authors suggest that melatonin supplementation improves mitochondrial dynamics.
The manuscript has been extensively modified, however there are some specific points to address.
- Why do the Authors use only the deep portion of Vastus Lateralis? I think that this selection could introduce a bias, because there could be a quantitative change of this fraction of the muscle in the different groups. I think it is more appropriate to analyze the whole muscle. If Authors want to study slow twitching muscles They could focus on soleus.
- Authors suggest that melatonin improves mitophagy because LC3bI and p62 are increased, however this is not precise. Melatonin has been described as a regulator of autophagy. Therefore, to demonstrate that cells increase mitophagy and not a general autophagic pathway, data should be supported by immunofluorescence analysis showing the co-localization of mitochondria and LC3.
- Authors report a quantification of MyHCs by means of SDS-PAGE. It is advisable to detect them by immunohistochemistry or immunofluorescence analyses to understand whether the changes observed in melatonin treated ZDF rats are ascribable to a quantitative fiber type switching. This will help to elucidated whether the increased NADH is due to a general increase of the mitochondria activity or to an increase of oxidative fibers.
- The same concept can be applied to CSA: in principle the increase of the mean fiber size is conflicting with the increase of slow-twitching fibers. The determination of the fiber type and the analysis of the CSA distribution are of help in explaining data. Moreover, recent data analyzing the metabolic profile of skeletal muscle fibers, show that fibers can be very plastic and adapt to different conditions, independently of MyHCs.
- The discussion is quite extensive
Some editing needed
Author Response
Response to reviewers’ comments
We thank the Reviewers for their comprehensive and insightful comments on our manuscript. Incorporation of these comments has certainly helped us to improve the quality of our manuscript considerably. Our responses to the individual comments and recommendations are as follows:
Reviewer # 1
Comment # 1
Why do the Authors use only the deep portion of Vastus Lateralis? I think that this selection could introduce a bias, because there could be a quantitative change of this fraction of the muscle in the different groups. I think it is more appropriate to analyze the whole muscle. If Authors want to study slow twitching muscles They could focus on soleus.
Response # 1
Our goal is to perform the study in a SKM with similar fast and slow twitch fibers in order to observe even small changes in caused by both pathology and treatment in all types of fibers. In addition, the vastus lateralis (VL) in rodents is one of the largest SKM involved in the movement. Therefore, it is the most similar to the rectus femoris in humans and the most translational study option. In addition, being a large muscle, it is expected to play an important role in SKM adaptation and metabolic plasticity regulation.
In the first instance, we studied the effect on the entire VL but after histological analysis we saw that in the superficial part of this muscle no significant changes were evident in any of the groups and we decided to focus the study on the deep part of the VL, where significant changes were found, so as not to make a global study of the VL and avoid masking small significant changes without losing the goal of this study (mixed twitch SKM). Also, deep VL in rats is the part in which all type fibers are present in similar percentage which was the goal of this study [1]. However, further detailed studies of the superficial VL and on different SKM are needed and are proposed in the revised version of the manuscript in the section "Discussion", Page 23 paragraph 1, line 828-832.
[1] Kohn TA, Myburgh KH. Regional specialization of rat quadriceps myosin heavy chain isoforms occurring in distal to proximal parts of middle and deep regions is not mirrored by citrate synthase activity. J Anat. 2007 Jan;210(1):8-18.
Comment # 2
Authors suggest that melatonin improves mitophagy because LC3bI and p62 are increased, however this is not precise. Melatonin has been described as a regulator of autophagy. Therefore, to demonstrate that cells increase mitophagy and not a general autophagic pathway, data should be supported by immunofluorescence analysis showing the co-localization of mitochondria and LC3.
Response # 2
Many thanks for this appreciation, we are completely agree. We have corrected this aspect by specifying that it is autophagy what is measured in this work (you can find it in yellow throughout the ms). The manuscript’s title has also been modified by exchanging only mitophagy by autophagy, becoming the final title “Melatonin Improves Skeletal Muscle Structure and Oxidative Phenotype by Regulating Mitochondrial Dynamics and Autophagy in Zücker Diabetic Fatty Rat”. Given the connection between autophagy and mitophagy, future studies are proposed to better discern the link between these processes of organellar dynamics and diabesity and how melatonin regulates them. You can find this in the revised ms in the section "Discussion", Page 20 paragraph 2, line 698-702.
Comment # 3
Authors report a quantification of MyHCs by means of SDS-PAGE. It is advisable to detect them by immunohistochemistry or immunofluorescence analyses to understand whether the changes observed in melatonin treated ZDF rats are ascribable to a quantitative fiber type switching. This will help to elucidated whether the increased NADH is due to a general increase of the mitochondria activity or to an increase of oxidative fibers.
Response # 3
We thank the reviewer for pointing this out. The reviewer brings up important points, which would be excellent research questions. However, the fiber-type specific effects of melatonin on fiber switching was deemed outside of the scope of the present study. The main goal of this study is to investigate the effects of melatonin on mitochondrial dynamic network and oxidative capacity. Additionally, since mitochondrial dysfunction plays a causal role in muscle inflexibility that occurs during obesity, we aimed at providing a preliminary insight into whether melatonin could restore obesity-induced fiber type transition and providing a basis for another study in the later context, which is ongoing. As reviewer comment, for understanding the fiber switching process, further detailed studies are needed and are proposed in the revised version of the manuscript in the section "Discussion", Page 22 paragraph 1, line 781-784.
Comment # 4
The same concept can be applied to CSA: in principle the increase of the mean fiber size is conflicting with the increase of slow-twitching fibers. The determination of the fiber type and the analysis of the CSA distribution are of help in explaining data. Moreover, recent data analyzing the metabolic profile of skeletal muscle fibers, show that fibers can be very plastic and adapt to different conditions, independently of MyHCs.
Response # 4
We thank the reviewer for pointing this out. We fully agree with what the reviewer mentioned. To address to this comment, Discussion section was improved and 2 new references were added to explain the plastic capacity of SKM to adapt to different conditions, independently of MyHCs characterization. Ms modifications corresponding to this comment are reflected in the "Discussion" section, Page 22 paragraph 1, line 776-784.
Comment # 5
The discussion is quite extensive.
Response # 5
Discussion was carefully revised and repeated information was removed to improve this section.
Reviewer 2 Report (New Reviewer)
Major and specific comments
In this study, entitled “Melatonin Improves Skeletal Muscle Structure and Oxidative phenotype by Regulating Mitochondrial Dynamics and Mitophagy in Zücker Diabetic Fatty Rat”, authors aimed to investigate the effects of melatonin on muscle structure and oxidative phenotype in diabetic fatty rats. They used Zücker diabetic fatty rats and their age-matched lean littermates as animal model, and their findings showed that mitochondrial fission and fusion was imbalanced in diabetic fatty rats. In addition, this study revealed that ATP production, SOD, and SIRT1 were downregulated, while nitro-oxidative stress was increased. Notably, melatonin treatments could restore TP production, SOD, and SIRT, as well as reduced oxidative stress and improved mitochondrial imbalance. Generally, this study has interests. Their experiments have been properly conducted, and the most of conclusions can be supported by their results. However, some issues still need to be clearly and further addressed. Here, several suggestions to improve the present manuscript are provided.
1. Since authors focused on exploring the effects of melatonin on diabetic muscle disorder, diabetic parameters in their animal model should be provided to support that these muscle disorders are attributable to diabetes (T2DM).
2. Did melatonin treatment promote muscular endurance or strength in these diabetic fatty rats?
3. In figure 2, p62 and LC3b/bI/bII are typical autophagic markers. However, there is no specific mitophagic markers such as Pink1/pT257-Pink1 and Parkin/pS65-Parkin.
4. Line 585, #P<0.05, but there is no # symbol in the figure 8.
5. Limitations in this study should be discussed.
Author Response
Response to reviewers’ comments
We thank the Reviewers for their comprehensive and insightful comments on our manuscript. Incorporation of these comments has certainly helped us to improve the quality of our manuscript considerably. Our responses to the individual comments and recommendations are as follows:
Reviewer # 2
Comment #1:
Since authors focused on exploring the effects of melatonin on diabetic muscle disorder, diabetic parameters in their animal model should be provided to support that these muscle disorders are attributable to diabetes (T2DM).
Response #1:
We thank the reviewer for this valuable comment. Parameters concerning T2DM analysis are previously published and they are attached here (see the table below). This our previous study from our team research group was mentioned in the “Introduction” section, Page 3 paragraph 2, line 107-108, and the new reference number [53]. “In addition to regulating circadian rhythms, it has antioxidant, anti-inflammatory, and energy balance regulating effects, which limit obesity, insulin resistance, hyperglycemia, and dyslipidemia [47–52], also in the same rat strain as showed in previous stud-ies of our research group [53].”
|
Group |
Fasting Glycemia (mg/dL) |
Glycated Hemoglobin (%) |
HOMA-IR |
|
C-ZL |
120 ± 29.8 |
3.89 ± 0.07 |
1.2 ± 0.12 |
|
M-ZL |
118 ± 12.5 |
3.92 ± 0.12 |
1.1 ± 0.15 |
|
C-ZDF |
460 ± 39.8 ### |
8.30 ± 0.48 ## |
9.3 ± 0.91 ### |
|
M-ZDF |
375 ± 34 * |
7.39 ± 0.32 * |
6.4 ± 0.87 ** |
# (ZL vs ZDF) à ## P<0.01, ###P<0.001
* (C vs M) à * P<0.05, ** P<0.01
Comment #2:
Did melatonin treatment promote muscular endurance or strength in these diabetic fatty rats?
Response #2:
We thank the reviewer for pointing this out. Any histological signs of muscle sarcopenia even damage was observed in ZDF rats so melatonin effects on sarcopenia in this murine model were not revealed. Many studies in sarcopenia and aging showed the effects of melatonin in muscle strength and endurance and a review was cited and a sentence was added highlighting the importance of this future study in the “Discussion” section, Page 22 paragraph 2, line 804-808.
Comment #3:
In figure 2, p62 and LC3b/bI/bII are typical autophagic markers. However, there is no specific mitophagic markers such as Pink1/pT257-Pink1 and Parkin/pS65-Parkin.
Response #3:
Many thanks for this appreciation, we are completely agree. We have corrected this aspect by specifying that it is autophagy what is measured in this work (you can find it in yellow throughout the ms). The manuscript’s title has also been modified by exchanging only mitophagy by autophagy, becoming the final title “Melatonin Improves Skeletal Muscle Structure and Oxidative Phenotype by Regulating Mitochondrial Dynamics and Autophagy in Zücker Diabetic Fatty Rat”. Given the connection between autophagy and mitophagy, future studies are proposed to better discern the link between these processes of organellar dynamics and diabesity and how melatonin regulates them. You can find this in the revised ms in the section "Discussion", Page 20 paragraph 2, line 698-702.
Comment #4
Line 585, #P<0.05, but there is no # symbol in the figure 8.
Response #3:
We thank the reviewer for pointing this out. We fully agree with what the reviewer mentioned. To address this comment, we have removed the #P<0.05 of the original manuscript at the legend of Figure 8.
Comment #5:
Limitations in this study should be discussed.
Response #5:
We are agree with this point and we thank the reviewer for his/her appreciation. Mitophagy and Fiber type characterization by immunofluorescence are the limitations of this study that are not performed. However, the fiber-type specific effects of melatonin on fiber switching was deemed outside of the scope of the present study. The main goal of this study is to investigate the effects of melatonin on mitochondrial dynamic network and oxidative capacity. As before comment, we corrected mitophagy reasults by autophagy one what is certainly studied in this work. Consequently mitophagy is the major limitation of this study and in the revised ms is mentioned toguether with other limitations (section "Discussion", Page 20 paragraph 2, line 698-702; Page 22 paragraph 1, line 776-784 ; Page 22 paragraph 2, line 804-808 ; Page 23 paragraph 1, line 828-832).
Round 2
Reviewer 1 Report (Previous Reviewer 3)
Authors answered and implemented the manuscript answering some of the questions raised. As pointed by the Authors, this work calls fo further investigation
Check minor spelling.
Reviewer 2 Report (New Reviewer)
The previous issues have been clearly addressed and the manuscript has been also properly improved. No further issues arose.
This manuscript is a resubmission of an earlier submission. The following is a list of the peer review reports and author responses from that submission.
Round 1
Reviewer 1 Report
I am not convinced at all by your answers.
A number of your results are simply aberrant and not at all in agreement with the bibliography.
The vastus lateralis is a rather glycolytic muscle in the rat. It has very few type I fibers (1 to 2%) and a very large majority of type IIb fibers (+60%) the rest being type IIa or IIx fibers. In this same muscle you find between 30 and 40% of type I fibers! This is not possible. This means that your measurements are bogus or that you have chosen areas that are not at all representative of this muscle.
There is a complete mismatch between your red oil staining and the quantification of intramuscular lipids that you show. Personally, I have never seen such an intensity of staining in a muscle, unless you choose very specific (and therefore non-representative) areas. However, the concentration of triglycerides that you find in your muscle homogenate is ridiculously low (about 400µg/g of tissue) when in principle it is more around 10mg/g.
Reviewer 2 Report
All concerns have been addressed.
Reviewer 3 Report
In the current version Authors improved some parts of the manuscript, however there are still some points that need further clarification and improvement.
Major points:
Since the title focuses on the correlation between skeletal muscle structure and oxidative phenotype, authors should report in the introduction what is the state of the art in this subject. There are some recent publications that report on the correlation between morphology and oxidative capacity in mouse models and humans.
The myosin chain isoform determination needs improvement. Authors state that there is a switch in fiber type, but there is not a precise determination (in lines 479-483, Authors write it has been impossible to determine the different subtypes). Actually, there are specific antibodies commercially available that can be used to detect the specific myosin isoforms. A correct classification of fiber types is important because type IIA fibers, despite been classified as fast, have an oxidative metabolism.
Moreover, it would be very interesting to understand if there is a variation of the oxidative capacity/mitochondrial activity in a fiber specific mode. Actually, recent results in aging mice demonstrated that IIA and IIX undergo a significant modification of the mitochondrial activity, while IIB fibers seem to be more resistant to modifications. This can have an important implication on the effects of melatonin on different skeletal muscles (here only a fast-contracting muscle is reported), because if there a general increase of the oxidative capacity, this will be beneficial for all muscles, while if the better oxidative capacity is fiber type dependent, melatonin activity will depend on the muscle type.
A similar consideration is valid for the lipid content.
Figure 6A ZDF-control reports a low-quality picture. Authors report that the fiber cross sectional area is reduced and that the morphology is altered, however in figure 6C and figure 7A there is not such an evident difference. Since the sections reported in Fig6A are from paraffin sectioning while in Fig6C and 7A from cryostat sectioning, it could be an artifact. I advise Authors to double-check the cross-sectional areas, and to report better representative images (also in Fig6C sample ZDF).
The analysis of the oxidative capacity (see figure 7B) is not correctly reported. It is advisable to perform the image analysis with imageJ/FiJi as described for the lipid staining. Images are converted to greyscale, the mean grey intensity values for each fiber analyzed will be calculated on a ROI drawn on the entire fiber. The values should be plot as arbitrary units.